# Non-Invasive Biomarkers in Saliva and Eye Infrared Thermography to Assess the Stress Response of Calves during Transport

**DOI:** 10.3390/ani13142311

**Published:** 2023-07-14

**Authors:** Mariana Caipira Lei, Luís Félix, Ricardo Cardoso, Sandra Mariza Monteiro, Severiano Silva, Carlos Venâncio

**Affiliations:** 1University Institute of Health Sciences (IUCS)—CESPU (IUCS-CESPU), 4585-116 Gandra, Portugal; marianalei96@gmail.com; 2Centre for the Research and Technology of Agro-Environment and Biological Sciences (CITAB), University of Trás-os-Montes and Alto Douro (UTAD), 5000-801 Vila Real, Portugal; ricardo19moc@gmail.com (R.C.); smonteir@utad.pt (S.M.M.); 3Inov4Agro, Institute for Innovation, Capacity Building and Sustainability of Agri-Food Production, University of Trás-os-Montes and Alto Douro, 5000-801 Vila Real, Portugal; 4Department of Biology and Environment, School of Life and Environmental Sciences, University of Trás-os-Montes and Alto Douro, 5000-801 Vila Real, Portugal; 5Department of Animal Science, School of Agrarian and Veterinary Sciences, University of Trás-os-Montes and Alto Douro, 5000-801 Vila Real, Portugal; ssilva@utad.pt; 6Veterinary and Animal Research Centre (CECAV), Associate Laboratory of Animal and Veterinary Sciences (AL4AnimalS), UTAD, Quinta de Prados, 5000-801 Vila Real, Portugal

**Keywords:** welfare, bovines, stress, oxidative stress, saliva, biomarkers, non-invasive techniques

## Abstract

**Simple Summary:**

Animal welfare and stress response assessment need valid animal-based indicators. Saliva has gained relevance as a non-invasive biological fluid that can be used to assess cortisol levels and different parameters of oxidative stress as potential sensitive indicators for evaluating animal welfare and health. The main objective of this work was to verify whether saliva and eye infrared thermography are accurate and efficient methods that can be used to detect transport-induced stress in calves. Our results show that transport increases salivary cortisol, oxidative status parameters, and eye temperatures. Furthermore, this study proves that the saliva of calves, as well as ocular temperature, suffers significant changes in its composition during the transport process, supporting saliva and infrared thermography as effective non-invasive methodologies to accurately assess cows’ stress status.

**Abstract:**

Animal transport is currently a stressful procedure. Therefore, animal-based indicators are needed for reliable and non-invasive welfare assessment. Saliva is a biospecimen with potential validity for the determination of cortisol and oxidative stress, although its use to assess calf welfare during transport has never been tested. Similarly, the applicability and reliability of infrared thermography to assess temperature change during calves’ transport have never been evaluated. These objectives were outlined following the known and growing need to identify non-invasive methodologies for stress assessment in bovines. This study was conducted on 20 calves of the Arouquesa autochthone breed, at about nine months of age, during their transport to slaughter. For each animal, saliva samples and thermographic images of the eye were collected at three time points: before transport, after transport, and at slaughter. The saliva was then processed to measure cortisol levels and oxidative stress parameters (reactive oxygen species, thiobarbituric acid reactive substance, carbonyls, and advanced oxidation protein products), and the images were analyzed using FLIR Tools+ software. There was an increase in cortisol concentration and oxidative stress parameters (reactive oxygen species, thiobarbituric acid reactive substance, carbonyls, and advanced oxidation protein products) in saliva after transport. An increase in eye temperature triggered by transport was also observed. The cortisol and eye temperature results at slaughter were returned to values similar to those before transport; however, the values of oxidative stress remained increased (mainly TBARS values). These non-invasive techniques seem to be reliable indicators of stress in bovine transport, and oxidative stress parameters in saliva may be a persistent marker for welfare assessment.

## 1. Introduction

Animal welfare is one of the main concerns in animal production, and it is imperative to develop animal-based measures to ensure stakeholders of the efficacy of its assessment [1]. One of the main concerns regarding animal welfare is their transport [2,3,4]. During transport, livestock is exposed to many psychological and physical stressors, such as unfamiliar noises and environments and commingling during handling, loading, and unloading [5]. Particularly in cattle, all these factors may trigger a stress response [2,3,4], which is more intense in calves [6].

It is widely accepted that compromised welfare, and the consequent physiological stress response, lead to rises in plasma cortisol levels. Thus, its assessment in the blood is the main biomarker of stress induction [7]. However, blood sampling is invasive, limiting its practical use [8]. Alternatively, saliva has gained prominence as a biological fluid in the assessment of cortisol since its collection is non-invasive [9]. In bovines, a correlation has been found between salivary cortisol and plasma cortisol concentrations [10,11]. Furthermore, the time lag that was thought to exist between evidence of changes in plasma and saliva cortisol concentrations seems to be reduced to only a few minutes in this species [8,10,11].

Even more, the stress response can trigger changes in various metabolic processes that can disturb homeostatic mechanisms between the production and neutralization of reactive oxygen species (ROS), favoring their production [12]. Recently, some studies have demonstrated that different oxidative stress parameters evaluated in saliva are sensitive indicators of animal welfare and health. These parameters have already been proven as markers of acute stress in sheep [13], mastitis in cows [14], weaning-induced stress in calves [15], and pain in pigs [16].

The temperature of the animal has also been considered as an indicator of a physiological response in the assessment of bovine welfare [17]. Since the temperature of the skin and extremities is very dependent on the peripherical blood flow, using infrared thermography (IRT), a non-invasive technique, it is possible to detect small variations in body temperature and measure changes in heat transfer and blood flow that occur during stressful situations [18,19,20]. Furthermore, this technique has allowed for the non-invasive analysis of ocular surface temperatures for stress assessment, allowing for a correlation with the animal’s internal body temperature [18,19]. Moreover, IRT can detect cattle with high temperatures during the antemortem period and predict the associated deterioration of meat quality parameters, such as dark, firm, and dry (DFD) meat [21,22].

The Arouquesa, as an example of bovine breeds in the Centre and North of Portugal, is generally transported for slaughter, preferably at around 9 months of age, a period of greater susceptibility to suffering from stress [23]. Therefore, this study proposes to evaluate the stress response in Arouquesa calves during transportation to the slaughterhouse, using animal-based parameters, through low-invasive techniques that evaluate the changes in ocular IRT images and oxidative stress parameters of saliva.

## 2. Materials and Methods

### 2.1. Animals and Samples Collection

In this study, 20 Arouquesa breed calves with an average age of 9 months (261.8 ± 33.4 days) and a live weight of 233.4 ± 40.7 kg were used. These animals were reared in traditional housing in their production area [23]. They were kept in pens of 2 to 5 each and a minimum space of 2.5 m^2^ per animal. None of the animals had ever been tied previously, and the animals were not accustomed to being handled or restrained. The transport took place between the farm and the slaughterhouse in Penafiel, using specific single-axle trucks suitable for transporting live animals, with a loading density range between 1.2 and 1.4 m^2^ per animal. The journey’s route, except at the beginning, was always the same, and the route was 62.5 ± 7 km over 264.2 ± 141.4 min. Sample collection took place in spring, and the ambient temperature on the days of transport was between 14 and 20 °C, with relative humidity usually under 80%, always under conditions of temperature–humidity index (THI) lower than 68, which is not conducive to inducing thermal stress [24]. In each animal, saliva samples and IRT eye images were collected at three sampling moments: before transport (BT), after transport (AT), and after a rest period (ARP) between 15 and 17 h. With regard to the last sampling time, IRT image collection occurred before stunning, with saliva collection immediately after stunning. During samplings, the animals were subjected to as little restraint as possible while ensuring the safety of all parties involved.

To capture the IRT ocular images, an infrared portable thermograph (FLIR F8, FLIR Systems AB, Stockholm, Sweden) was used with a detector of 320 × 240 pixels and adopting an emissivity of 0.98. The animals were sheltered from direct solar radiation, air flows, and rain. The camera was positioned perpendicularly to the eye to be evaluated at approximately the same distance and never closer than 0.5 m, as shown in Figure 1.

The collection of saliva samples was obtained with a cotton ball that was inserted, with the help of tweezers, into the animal’s mouth through the labial commissure when in movement on the way through the passageway for loading and in the passageway upon arrival at the slaughterhouse. When related to saliva, the cotton was stored in screw-top containers that were correctly identified. After collection, the containers were stored at refrigeration temperatures until the samples’ processing in the lab. The cotton from each animal was centrifugated at 4000 rpm for 10 min using a Janetzki T150 (Janetzki KG Heinz, Leipzig, Germany), and the obtained sample was portioned and stored at −20 °C until its use in analysis.

### 2.2. Cortisol Measurement

For the analyses of the salivary cortisol, the samples were submitted to an extraction process with ether based on the methodology of Zeugswetter et al. [25]. First, to 100 μL of the sample, 1 mL of diethyl ether was added under slow stirring using an Edmund Buhler KL2 shaker. The agitation continued for 24 h to ensure that all cortisol would be extracted from the cotton. The obtained samples were frozen to make the collection of the organic phase to new tubes easier. After, diethyl ether was removed in a rotary evaporator (Labconco CentriVap) at 45 °C for about 1 h, and 200 μL of phosphate-buffered saline (PBS) was added to each microtube. To concretely determine the concentration of cortisol in saliva, a Salivary Cortisol ELISA Kit (Salimetrics^®^, LLC 1-3002) was used according to the manufacturer’s instructions. To obtain the concentrations of cortisol (ng/mL), the results were read at 450 nm, later corrected to 490 nm to discard the interferences, in a PowerWave XS2 (Bio-TeK^®^ instruments, EUA) microplate reader.

### 2.3. Measurement of the Oxidative Stress Markers

#### 2.3.1. Thiobarbituric Acid Reactive Substance (TBARS)

This test was performed after an adaptation of a method already described [26]. Into each well of a 96-well microplate, the following were added: 40 μL of the sample; 70 μL of water; and 70 μL of phosphate buffer, 50 mM at pH 7.4. After, 10 μL of butylated hydroxytoluene (BHT) 1 mM, 75 μL of thiobarbituric acid (TBA) 1.3% in NaOH 0.3%, and 50 μL of trichloroacetic acid (TCA) 50% were added. The preparation was incubated at 60 °C for 40 min, the linking of TBA with MDA was measured at a wavelength of 530 nm, and a wavelength of 600 nm was used to remove the absorption of non-specific TBA-reactive substances. The results were expressed according to MDA (0–0.025 mM) standard curve in (MDA) mM/mg of protein.

#### 2.3.2. 2′-7′-Dichlorofluorescein Diacetate Staining

This test was performed after an adaptation of a method already described [27]. In each well of a 96-well microplate were added: 40 μL of the sample, 100 μL of phosphate buffer at pH 7.4, 8.3 μL of 2′-7′-dichlorofluorescein diacetate (DCFH-DA) 10 mg/mL (in Dimethyl sulfoxide—DMSO). The mixture was incubated at 37 °C for 30 min. After incubation, samples were read in the fluorimeter (Cary Eclipse from Varian company, Palo Alto, California, United States of America) at an excitation of 485 nm and emission of 530 nm. Dichlorofluorescein (DCF) was used as standard (stock in DMSO), with the maximum being 0.25 mM. The results were expressed in μmol DCF/mg of protein.

#### 2.3.3. Protein Carbonyls

This test was performed after an adaptation of a method already described [28]. A total of 40 microliters of sample and 40 μL of DNPH (2,4-dinitrophenylhydrazine) at 10 mM were added to each of the wells in a 96-well microplate. This mixture was incubated in the dark at room temperature for 10 min, and then 20 μL of 6 M NaOH was added. The reading was made at 450 nm after another 10 min of incubation. The results were expressed in (DNPH) mM/mg of protein considering the molar extinction coefficient of DNPH 22.308 mM^−1^cm^−1^.

#### 2.3.4. Advanced Oxidation Protein Products (AOPP)

This test was performed to evaluate the oxidative damage of proteins based on a method previously described [13]. In each of the 96 wells of a microplate, 40 μL of the sample and 25 μL of 50% acetic acid were added. At that moment, one reading was taken at 340 nm. Then, 200 μL of 60 mM KI was added. After 5 min, a new reading was taken, and the difference in absorbances was quantified based on a chloramine-T (0–0.2 mM) standard curve, and the results were expressed as (Chloramine-T) mM/mg protein.

### 2.4. Eye IRT Data Process

Using the eye IRT images and FLIR Tools software, the maximum and mean eye temperature data were determined. To ensure exactitude in maximum eye temperature determination, the ellipse tool of the software was used, as well as an ellipse fitted to the animal eye [29]. Maximum (IRTmax) and mean (IRTmean) temperatures were obtained for each selected image (Figure 1).

### 2.5. Statistical Analysis

Statistical analysis was performed using GraphPad Prism software (version 8), and the significance level was set to *p* < 0.05. The outliers were identified and removed using the ROUT method with a Q value of 1%. Animals without 3 measurements/repeats were also excluded.

The normality of each parameter was evaluated with the Shapiro–Wilk test, attending to the small size of the sample. For variables presenting a normal distribution, homogeneity of means was tested with the Brown–Forsythe test. Comparisons between groups were created using the one-way ANOVA. The differences between groups were determined with the Tukey test. These data have been presented as means and standard deviations. For variables with *p*-value ≤ 0.05 (not normally distributed), the non-parametric test analysis—Friedman test—was performed. The differences between groups were assessed with the Dunn test. These data have been presented as median and interquartile ranges.

## 3. Results and Discussion

This work aimed to verify whether saliva and infrared thermography of the eye could be considered efficient non-invasive methods to detect stress responses in Arouquesa calves after their transport to a slaughterhouse. To the best of the authors’ knowledge, this is the first study to present a number of oxidative stress biomarkers able to detect a stress response triggered by transport in this animal model. In addition, eye infrared thermography was validated as an accurate method that could be used to detect transport stress.

The results show statistically significant differences in cortisol between the groups (X(2) = 0.4506, *p* = 0.0004), confirming that transport is a stressful event and that cortisol reflects the stress levels efficiently (Figure 2). Indeed, there are highly significant differences between the cortisol levels before and after transport (BT vs. AT, *p* = 0.0008), and very significant differences between the moment after transport and after a rest period (AT vs. ARP, *p* = 0.0048). Cortisol levels in blood are commonly reported as the parameter most affected by transport [30]. The increases in salivary cortisol were also confirmed after a stress stimulation in cows [8,11]. The sharp rise of salivary cortisol levels observed in Arouquesa calves from pre- to after-transport moment (0.67 ± 0.49 ng/mL to 1.98 ± 1.34 ng/mL) suggests that this increase was triggered by transport-induced stress. In mammals, adrenaline release is the first response to a stressful situation, followed by a long-term response ensured by the release of glucocorticoids, mainly blood cortisol [5]. Thus, this hypothesis is supported by the results, which show an increase in saliva cortisol levels after transport. The cortisol values were, however, restored (0.86 ± 0.43 ng/mL) after 16–18 h of rest, when the source of stress (transport) was absent. This agrees with the study performed by Cook et al. [31], in which salivary cortisol levels were restored after an 8 h rest period. However, it should be noted that in long-distance transport, recovery to pre-transport values may take 5 to 16 days due to exhaustion, dysregulation of eating and water routines [32,33], and a lack of prolonged release of cortisol due to negative feedback control of the hypothalamic–pituitary–adrenal axis after an initial acute response to handling and loading [31]. These findings suggest that the negative impacts of high levels of cortisol can remain for several days before returning to normal values. It is crucial to note that standard farm procedures and even non-aversive environments can raise cortisol levels, potentially affecting sampling results. Thus, it is essential to continue to identify reliable non-invasive markers for stress assessment [8,11,32].

Normal metabolism implies the conversion of oxygen into ROS during ATP production. Situations that demand more energy, and consequently oxygen, result in a higher production of ROS, mainly if there is no prior adaptation to the demanding situation [34]. In the present study, the observed ROS concentrations agree with the evolution tendency of cortisol levels (Figure 3). There was an increase associated with transport (from 0.01 ± 0.01 mM of DCF/mg of protein BT to 0.07 ± 0.04 mM of DCF/mg of protein AT) and then a decrease after a rest period (from 0.07 ± 0.04 mM of DCF/mg of protein AT to 0.04 ± 0.02 mM of DCF/mg of protein ARP). Statistically significant differences were found (X(2) = 0.5742, *p* < 0.0001) between all pairs of groups: BT vs. AT, *p* < 0.0001; BT vs. ARP, *p* = 0.0001; AT vs. ARP, *p* = 0.0038, proving that these events are associated and that transport truly increases ROS production. Short-term stress (such as our 4.5 h transport), the periods of fasting that cattle may experience before arrival, and exhaustion from muscle fatigue from trying to maintain balance worsen ROS production [5,34]. It is also important to remember that, during transit, cattle are also exposed to increased levels of exogenous ROS from the environment, such as exhaust fumes, UV light, and pollutants [5]. The results of the present study agree with the results of Urban-Chmiel et al. (2009), who reported an increase in ROS production post-transit in bovines [35], and of Piccione et al. (2013), who reported that the increase found in sheep serum remained at 12, 24, and 48 h post-transit [36]. The persistence of the high concentrations is in agreement with the high values, which, in the present study, remained after 18 h of resting.

The overproduction of oxidant compounds, such as ROS, relative to antioxidant compounds establishes an oxidative stress status [37]. The surplus causes oxidative modifications in proteins, lipids, and nucleic acids, whose metabolites can be used as oxidative stress biomarkers [5].

Lipid peroxidation is the best-described consequence of ROS generation [33,38], and the intensity of this phenomenon can be evaluated by the determination of thiobarbituric acid reactive substances (TBARS) [33,39]. In the present study (Figure 3), TBARS evolution suggests that transport induces an increase in lipid peroxidation (BT vs. AT, *p* = 0.0007) with no recovery after 16–18 h of resting (0.008 ± 0.004 mM of MDA/mg of protein BT, 0.02 ± 0.002 mM of MDA/mg of protein AT, and 0.02 ± 0.002 mM of MDA/mg of protein ARP). Although the differences between AT and ARP were not statistically significant (*p* = 0.10), the results suggest that recovery from an event as stressful as transport in terms of lipidic peroxidation may take some time. Wernicki et al. [33] verified that TBARS increased significantly until day 3 after transport. The recovery of the values to those before transport took 22 days.

Other targets of oxidative compounds are proteins [38]. Their exposure to free radicals creates advanced oxidation protein products (AOPP), whose accumulation has been shown to promote acute states of inflammation [39]. According to Celi, P. (2010), AOPP might also be a by-product of neutrophil activation, justifying its association with inflammation [39]. AOPP showed statistically highly significant differences between groups (X(2) = 0.5018, *p* < 0.0001). Differences were found between the AOPP measurements before and after transport and between the moment before transport and after a rest period (BT vs. ARP, *p* = 0.0003), proving that a real association exists between transport and AOPP levels. Along with this study’s significant increase in AOPP concentration after animal transport (Figure 3), there were damaged proteins (0.24 ± 0.09 mM of chloramine-T/mg of protein BT to 0.51 ± 0.21 mM of chloramine-T/mg of protein AT, BT vs. AT, *p* < 0.0001). The already observed correlation between salivary cortisol and AOPP in calves (at weaning and grouping) supports the hypothesis that transport-induced oxidative stress can be triggered by heightened cortisol levels [15]. The decrease in the AOPP concentration verified after the rest period (0.51 ± 0.21 mM of chloramine-T/mg of protein after transport to 0.44 ± 0.14 mM of chloramine-T/mg of protein after a rest period) agrees with the results obtained by Rubio et al.; two days after weaning, calves AOPP salivary levels were lower than those found at the weaning moment [15]. This is probably because after the extinction of the acute stress and oxidative stress, AOPP starts to return to regular values.

Another frequent modification of proteins induced by ROS is protein carbonylation [40]. In this study (Figure 3), the protein carbonyls concentration increased in response to transport-induced stress (0.05 ± 0.03 mM of DNPH/mg of protein BT to 0.09 ± 0.08 mM of DNPH/mg of protein AT) as a consequence of the induced oxidative stress. These findings agree with those found by Marco-Ramell et al., which showed that cows living in challenging environments, with food restriction and far from human contact, presented increased levels of protein carbonyls compared to those living in low-challenging systems [41]. Furthermore, high levels of protein carbonyls were also associated with endometritis and lameness in dairy cows [42,43]. Although no studies were found on the evolution of this parameter in ruminants after the resolution of an acute stress factor, Herasymets et al. verified that the high concentrations of 2,4-DNPH observed after simulating acute hepatitis in rats would progressively reduce after the administration of corrective compounds [44]. This tendency agrees with the decrease in DNPH values observed from the moment after transport to after a rest period (0.09 ± 0.08 mM of DNPH/mg of protein AT to 0.08 ± 0.05 mM of DNPH/mg of protein ARP), suggesting that after the episode of acute stress (transport), protein carbonylation starts to decrease. However, in the present study, no statistically significant differences between groups (X(2) = 0.1399, *p* = 0.1171) were reported. This can be due to the biospecimen used, as its composition in proteins and lipids seems to affect the type and intensity of oxidation processes, as verified by Xie et al. [45]. Also, as in Herasymets et al.’s study, it is possible that more days after transport would be needed to find statistically significant differences between these two moments [44].

Statistically significant differences were found between the groups’ mean IRT temperature (X(2) = 0.3214, *p* = 0.0010). Statistically significant differences were found between the mean temperature before and after transport (BT vs. AT, *p* = 0.0041), as well as highly significant differences between the moment after transport and after a rest period (AT vs. ARP, *p* = 0.0010). Regarding the maximum IRT temperature, statistically significant differences were found (X(2) = 0.4159, *p* = 0.0001) between the measurements before and after transport (BT vs. AT, *p* = 0.0245), and highly significant differences were found between the moment after transport and after a rest period (AT vs. ARP, *p* < 0.0001) (Figure 4). During a stressful situation, there is an increase in body temperature and changes in blood flow that can be detected using IRT. Several studies using different body regions have already proved this technology’s utility in evaluating stress (Reviewed by [18]). In this study, a window including the eye and the lacrimal caruncle was used. The orbital region has a high vascularization, from the facial and infraorbital arteries, innervated by sympathetic fibers from the facial nerve and allowing them to respond to stressful and harmful stimuli. These fibers are sensitive to epinephrine and norepinephrine, which increase the heart rate, vasoconstriction, and blood pressure, thus reducing the heat exchange rate and functioning as a local thermoregulating mechanism [20,46,47]. The eye also has the advantage of allowing measurements without the interference of fur, hair, or dirt. The mean and maximum IRT temperatures evolved similarly and followed the trend of the cortisol concentration. Before transport, the mean temperature was 33.11 ± 1.27 °C and the maximum temperature was 36.05 ± 1.10 °C; increasing to 34.36 ± 0.99 °C and 37.07 ± 0.78 °C, respectively, immediately after transport; and decreasing to a mean temperature of 33.13 ± 1.21 °C and maximum of 35.48 ± 0.79 °C after a rest period. Eye temperature changes in response to stress or pain are not fully understood. Most previous studies suggest a drop in maximum temperature after a stressful or painful stimulus, while others report an increase. Stewart, M. stated that this last aspect may have failed to detect an initial drop in eye temperature because of infrequent sampling. The drop in eye temperature is a sympathetically mediated response due to vasoconstriction triggered by the release of adrenaline, while parasympathetic activity lowers cardiac output and blood pressure, resulting in vasodilation, which could cause an increase in eye temperature. However, the nature of the pain or stress may influence the response, as found by Stewart, M., who observed an increase in response to castration (deep visceral pain) and a lower increase to disbudding (somatic pain) [47]. Curiously, startling may also increase one’s eye temperature, as observed in humans in response to sudden loud noises [48]. Additionally, studies suggest that animals may not display signs of stress unless they have cognitive awareness of it. Cows showed increased eye temperature after social isolation and a second catheterization, indicating that anticipation and cognitive components may be perceived stressors [20]. There is also no consensus regarding rectal temperature. Some studies already verified an increase in the temperature of young calves and steers after road transport [32,49,50]. Some studies have found a decrease in rectal temperature during stressful situations, which may be due to a decrease in visceral activity [51] or to long-distance travel, when fasting and habituation lead to a decrease in body temperature [52]. Cow body temperature and IRT in the eye can also increase in a situation of thermal stress, particularly reflecting THI values above 70 [53,54], which was not the case in this work, where the weather conditions were very mild. The recovery of the temperature to values near those found pre-transport in this study (mean temperature of 33.13 ± 1.21 °C and maximum of 35.48 ± 0.79 °C) is in agreement with several studies that reported similar results after exposing a ruminant to stress (transport or other); the recovery of body temperature to pre-stress values is relatively quick [52,55]. These findings show once again that transport triggers stress and that IRT detects it efficiently. Stewart et al. [20] also verified that IRT of the eye is a valid method that can be used to assess animal stress induced by catheterization, a procedure similarly stressful to transport. Valera et al. also confirmed, in competition horses, that the use of these two parameters, IRT eye temperature and salivary cortisol, are efficient when used to evaluate stress [56], since Cook et al. already verified that a significant correlation exists between salivary or plasma cortisol and eye temperature [47].

## 4. Conclusions

This study found that transport of Arouquesa calves induces a stress response reflected in changes in salivary cortisol levels, salivary oxidative parameters, and eye temperature. We verified that salivary ROS, TBARS, and AOPP are accurate biomarkers that can be used to detect the above-mentioned stress. Saliva analyses and IRT are adequate methodologies that can be used to assess transport stress.

These findings are important steps in the search for non-invasive methodologies to detect transport stress. IRT could be integrated at different control points in an automated system, allowing for the early detection of severely distressed animals. IRT and salivary oxidative stress biomarkers can help ensure that animals are not exposed to stressful situations. Further studies should investigate the recovery of these variables after transport and their impact on meat quality.

## Figures and Tables

**Figure 1 animals-13-02311-f001:**
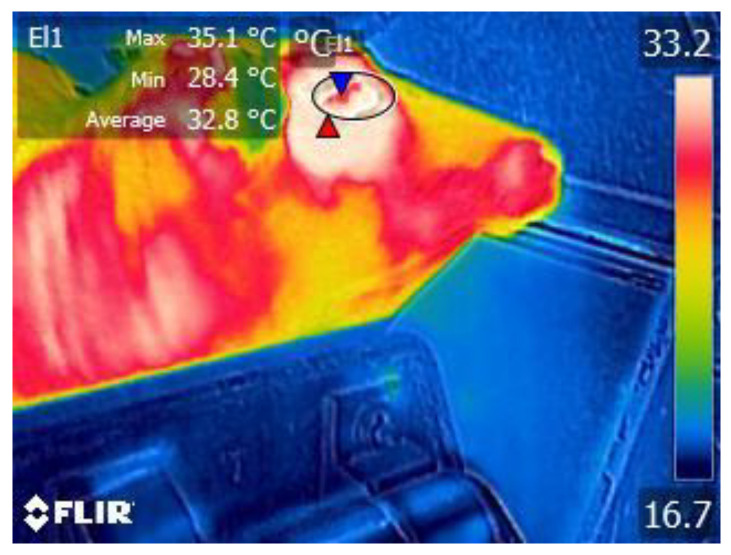
Representative IRT image processing of an Arouquesa calve’s right eye.

**Figure 2 animals-13-02311-f002:**
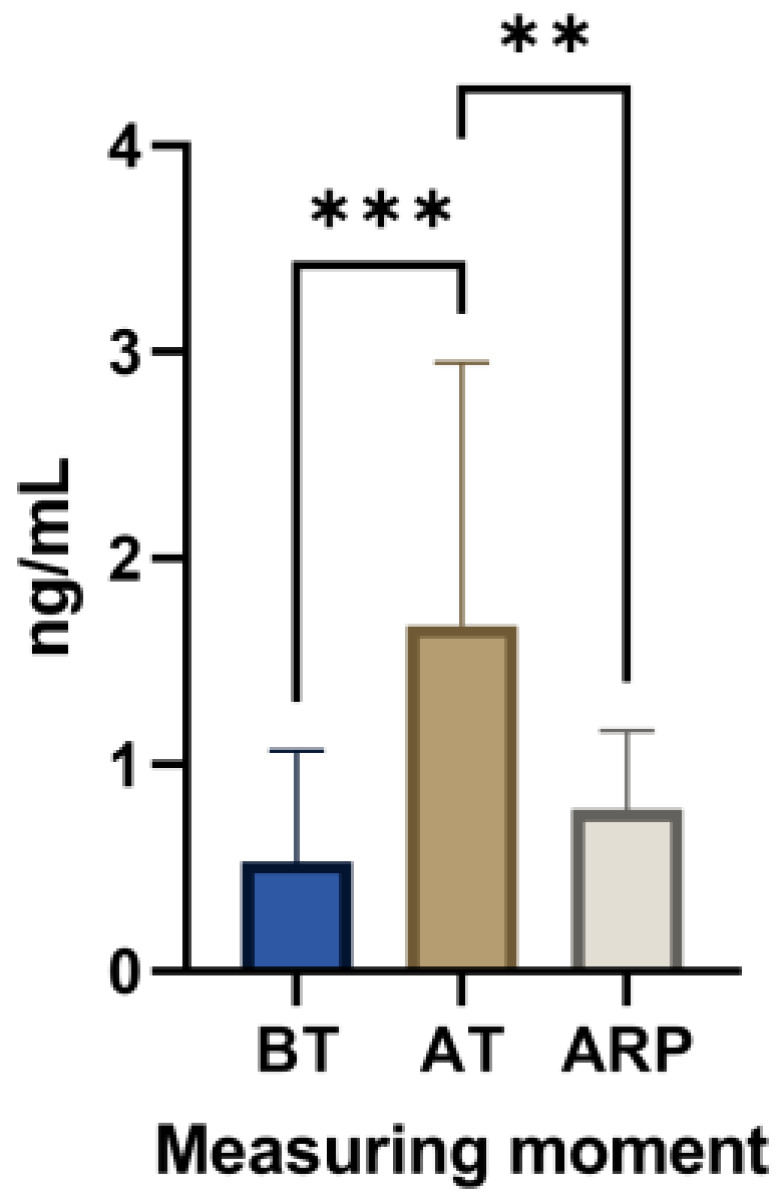
Salivary cortisol levels of Arouquesa calves before transport (BT), after transport (AT), and after a resting period (ARP) between transport and slaughter. ** *p* ≤ 0.01; *** *p* ≤ 0.001.

**Figure 3 animals-13-02311-f003:**
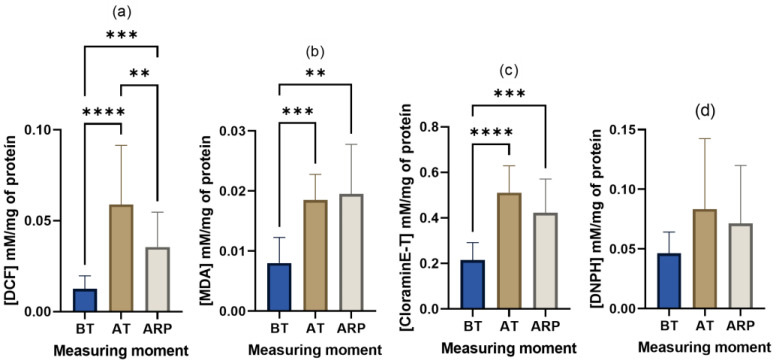
Results of (**a**) salivary reactive oxygen species (ROS), (**b**) salivary thiobarbituric acid reactive substance (TBARS), (**c**) salivary advanced oxidation protein products (AOPP), and (**d**) salivary carbonyls of Arouquesa calves before transport (BT), after transport (AT), and after a resting period (ARP) between transport and slaughter. ** *p* ≤ 0.01; *** *p* ≤ 0.001; **** *p* ≤ 0.0001.

**Figure 4 animals-13-02311-f004:**
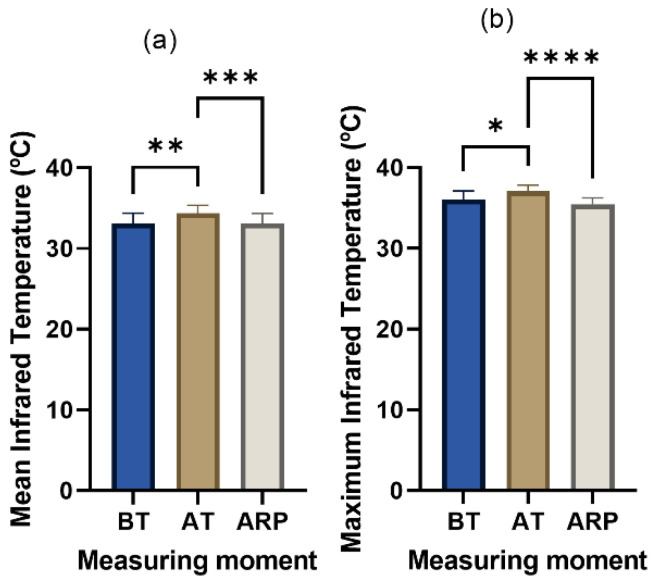
Results of the (**a**) mean and (**b**) maximum infrared thermography temperature of one eye and respective lacrimal caruncule of Arouquesa calves before transport (BT), after transport (AT), and after a resting period (ARP) between transport and slaughter. * *p* ≤ 0.05; ** *p* ≤ 0.01; *** *p* ≤ 0.001; **** *p* ≤ 0.0001.

## Data Availability

The data presented in this study are available upon request from the corresponding author. The data are not publicly available to preserve the privacy of the data.

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
