# Peer review of "Non-Invasive Biomarkers in Saliva and Eye Infrared Thermography to Assess the Stress Response of Calves during Transport"

_animals, 2023, doi:10.3390/ani13142311_

Round 1

Reviewer 1 Report

Comments

1. This work aimed to verify whether saliva and eye infrared thermography could be 196

considered efficient non-invasive methods to detect stress response in Arouquesa calves 197

after transport to slaughterhouse. But the eye infrared thermography was not included in the title.

2. In results and discussion, Cortisol level is a improtant stress index of transport, but the author does not explain clear that most os the references were on blood cortisol. It is need to be discussion clear the cortisol levels were from what samples.

3. In this study, it is necessary to measure the blood corticol and rectal temperature, and reveal the correlation between blood corticol and saliva corticol in the paper.

4. the eyes temperature is easy to be impacted by the environment temperature, not only by the stress. In the paper, there is no information on environment temperature before or after transport.

5. In figure 1, the temperature of eyes (max, mini and average) of the pics were quite different, what is that reason? Is it bacause the distance of the eyes to the thermograph?

6.  On line 97, 62.5 ± 7 km during 264.2 ± 141.4 minutes, it is a huge variation of transport time, so was the data from this tranport was persuasive for a scientific paper?

Comments

1. This work aimed to verify whether saliva and eye infrared thermography could be 196

considered efficient non-invasive methods to detect stress response in Arouquesa calves 197

after transport to slaughterhouse. But the eye infrared thermography was not included in the title.

2. In results and discussion, Cortisol level is a improtant stress index of transport, but the author does not explain clear that most os the references were on blood cortisol. It is need to be discussion clear the cortisol levels were from what samples.

3. In this study, it is necessary to measure the blood corticol and rectal temperature, and reveal the correlation between blood corticol and saliva corticol in the paper.

4. the eyes temperature is easy to be impacted by the environment temperature, not only by the stress. In the paper, there is no information on environment temperature before or after transport.

5. In figure 1, the temperature of eyes (max, mini and average) of the pics were quite different, what is that reason? Is it bacause the distance of the eyes to the thermograph?

6.  On line 97, 62.5 ± 7 km during 264.2 ± 141.4 minutes, it is a huge variation of transport time, so was the data from this tranport was persuasive for a scientific paper?

Author Response

Responses to the comments of Reviewer #1

  1. This work aimed to verify whether saliva and eye infrared thermography could be considered efficient non-invasive methods to detect stress response in Arouquesa calves after transport to slaughterhouse. But the eye infrared thermography was not included in the title.

Response:  The authors appreciate the comment by the reviewer and have changed the title accordingly.

  1. In results and discussion, Cortisol level is a improtant stress index of transport, but the author does not explain clear that most os the references were on blood cortisol. It is need to be discussion clear the cortisol levels were from what samples.

Response:  The authors acknowledge this limitation and have made corrections to the text taking in consideration the issue raised by the reviewer.

  1. In this study, it is necessary to measure the blood corticol and rectal temperature, and reveal the correlation between blood corticol and saliva corticol in the paper.

Response:   The authors recognize that rectal temperature and plasma cortisol data would be important to support the stress response that occurred during transport, however, to simplify the procedures with the animals, it was not planned to collect blood samples or rectal temperature. Because we were aware that cortisol in saliva is a sufficiently robust marker to support a possible stress response (10.1186/s13028-014-0061-3; 10.5713/ajas.18.0151). 

  1. The eyes temperature is easy to be impacted by the environment temperature, not only by the stress. In the paper, there is no information on environment temperature before or after transport.

Response:  The reviewer is right to point out the effect of environmental temperature and its potential influence on eye temperature. However, in the spring months, the region where the data collection took place, has a very mild climate. The conditions of temperature ranging between 14 and 20 ºC and relative humidity usually under 80%, always under conditions of temperature-humidity index (THI) lower than 68. Overall, these conditions are not conducive to inducing thermal stress as previously described (10.3168/jds.2021-21164). Moreover, we believe that under these environmental conditions the influence on eye temperature will be minimal (10.1007/s00484-018-01666-x; 10.5713/ajas.19.0762). To clarify this issue, transportation conditions were included in the revised manuscript.

  1. In figure 1, the temperature of eyes (max, mini and average) of the pics were quite different, what is that reason? Is it bacause the distance of the eyes to the thermograph?

Response:  The authors appreciate the comment raised. In fact, Figure 1 is a set of two thermographic images, one after transport (AT) and the other after the rest period (ARP). However, the authors apologize for choosing the wrong images. All images were taken at approximately the same distance and never closer than 0.5 m and only slight variations were observed as shown in Figure 4. To clarify this issue, these indications were included in the revised manuscript.

  1. On line 97, 62.5 ± 7 km during 264.2 ± 141.4 minutes, it is a huge variation of transport time, so was the data from this tranport was persuasive for a scientific paper?

Response:  In fact, the variability recorded in the duration of transport is the time that the animals remained in the vehicle, which includes the waiting period to leave after loading, the duration of the journey on a winding road with steep slopes, any periods of stoppage, as well as waiting to unload due to occupancy of the quay at the slaughterhouse. The authors understand the observation made by the reviewer, but the variability presented is representative of the existing reality and should be one of the factors to be controlled in future works to control the stress response observed in these animals and its possible consequences.

Reviewer 2 Report

In the beef cattle industry, transportation is indeed very common, transportation will lead to stress, reduced immunity, increased BRD rate, increased mortality, reduced meat quality, etc., There are no similar studies exist, the use of non-invasive technical means in the investigation of whether animals are in a state of stress, the novelty of this study is high.

However, I have 2 simple questions for editors and authors to consider:

1.     As a methodological pioneer, I think the number of 20 cattle is too small, is it representative, and can it be generalized in future ?

2.     Since you have conducted the detection of indicators at three time points, why did you not set a treatment group to alleviate transportation stress when designing the experiment, and test the results have you reduced these stress indicators? In this way, the method is more convincing.

Author Response

Responses to the comments of Reviewer #2

  1. As a methodological pioneer, I think the number of 20 cattle is too small, is it representative, and can it be generalized in future?

Response:  The authors appreciate the question raised by the reviewer. Although not described in the manuscript, based on the values shown in different related literature where cortisol increased after transportation, an alpha of 0.05, and a power of 0.80, and assuming a medium effect in G*Power software, authors would need to collect observations from 9 animals to detect a significant treatment effect. Yet, this number was increased to 20 cattle. As such, we believe that the results can be regarded are representative for the population/transportation of animals under similar conditions.

  1. Since you have conducted the detection of indicators at three time points, why did you not set a treatment group to alleviate transportation stress when designing the experiment, and test the results have you reduced these stress indicators? In this way, the method is more convincing.

Response: The authors appreciate the comment raised by the reviewer which will be considered for further studies. In fact, the objective of this pioneer study was to confirm the occurrence of stress during transportation of these animals and the implementation of non-invasive biomarkers to test this assumption, rather than test an effective treatment to reduce or alleviate the transportation-induced stress.

Reviewer 3 Report

Dear authors, your maniscript is well written. Nevertheless, I do have some suggestions to improve the manuscript for better understanding. 

Line 127: please add space beteen „200 µL“ and „of“

Line 207: please delete space in „p= 0.0048“

Line 247: I believe the two publications, cited in the sentence are missing in the reference list.

Lines 281 and 282: please delete space between cholamine-T /mg

Line 301: change „start to decrease“ into „starts to decrease“

Line 317: please delete space between p= 0.0010

Line 332: please delete space between ± 1.27 but add space betwee 1.27°C (same in the lines 333 and 355)

Lines 371-374: Maybe add these sentences into the conclusions part and delete here.

·        The materials and methods part should include more information about animals and farms. Did all calves came from the same farm? Were they of different sex? Have all calves been handeled equally before testing?

·        Why did you not use a mixed model, which includes several effects such as sex, live weight, age, climatic situation during transport (were some animals heat stressed during transport while others were not, different seasons?) and distance from farm to slaughterhouse (if animals came from different farms)?

·        Why not including a table representing the most relevant results?

Some sentences are very long e.g. lines 140-143 or lines 332-335. It would be easier to understand, if you could write two shorter sentences instead.

Line 65: please change „specie“ in „species“

Line 169: please change „5 minutes after“ into „5 minutes later“ or „after 5 minutes“

Line 190: please change „…presented as mean and standard deviation“ into „…presented as means and standard deviations“

Line 204: maybe change „that stress“ into „the stress“

Line 251: maybe change „that in the present study…“ into „which in the present study…“

Line 356: maybe change „…reported that after….“ Into „…reported similar results after…“

Author Response

Responses to the comments of Reviewer #3

  1. Line 127: please add space beteen „200 µL“ and „of“

Response: Changes were made accordingly.

  1. Line 207: please delete space in „p= 0.0048“

Response:  Changes were made accordingly.

  1. Line 247: I believe the two publications, cited in the sentence are missing in the reference list.

Response:  The references were checked.

  1. Lines 281 and 282: please delete space between cholamine-T /mg

Response:  Changes were made accordingly.

  1. Line 301: change „start to decrease“ into „starts to decrease“

Response:  Changes were made accordingly.

  1. Line 317: please delete space between p= 0.0010

Response:  Changes were made accordingly.

  1. Line 332: please delete space between ± 1.27 but add space betwee 1.27°C (same in the lines 333 and 355)

Response:  Changes were made accordingly.

  1. Lines 371-374: Maybe add these sentences into the conclusions part and delete here.

Response:  These sentences were removed as the same ideas were already included in the conclusion.

  1. The materials and methods part should include more information about animals and farms. Did all calves came from the same farm? Were they of different sex? Have all calves been handeled equally before testing?

Response:  Regarding this issue, and in order to minimize bias, all animals were subjected to the same procedure. The animals were stopped in route for loading and on arrival at the slaughterhouse to collect saliva. The animals were reared in traditional housing in their production area with their centre in the municipality of Cinfães and on farms at relatively close distances. The route to the slaughterhouse in Penafiel was the same. The animals used were of both sexes, but this was not taken into consideration because no sex-specific effect on cortisol release has been described at these ages (10.2527/jas1984.592376x; 10.2527/2003.81112847x).

  1. Why did you not use a mixed model, which includes several effects such as sex, live weight, age, climatic situation during transport (were some animals heat stressed during transport while others were not, different seasons?) and distance from farm to slaughterhouse (if animals came from different farms)?

Response:  The factors and analysis approach can be important for a better understanding of the factors that may be triggering stress during transport. However, in this work we intend to confirm the suspicion that calves of the Arouquesa breed during the normal transportation process to slaughter could be subject to a stress response and to use them as a model to test the evaluation methodology and the oxidative stress parameters in saliva as markers of this response.  Sex was not taken into consideration because this factor has no effect on cortisol release at these ages (10.2527/jas1984.592376x; 10.2527/2003.81112847x). The age and size are also not considered relevant considering that the variability observed is normal within the destination in which they are inserted, the production of veal. Sample collection took place in Spring and the temperature and relative humidity on the days of transport were between 14 and 20 ºC and under 80% respectively. Thus, these conditions are not conducive to inducing thermal stress as previously described, with possible changes for temperatures above 25°C (10.3168/jds.2019-17929; 10.3168/jds.2021-21164). The animals were reared in traditional housing in their production area with their centre in the municipality of Cinfães and at relatively close distances, but the route to the slaughterhouse was always the same.

  1. Why not including a table representing the most relevant results?

Response:  The authors preferred to show the most relevant data in graphs as they are better for perceiving trends and making comparisons and predictions between groups.

  1. Some sentences are very long e.g. lines 140-143 or lines 332-335. It would be easier to understand, if you could write two shorter sentences instead.

Response:  We apologize but the authors believe that the sentences are not very long and easy to understand. Changing this would change the interpretation of the sentences.

  1. Line 65: please change „specie“ in „species“

Response:  Changes were made accordingly.

  1. Line 169: please change „5 minutes after“ into „5 minutes later“ or „after 5 minutes“

Response:  Changes were made accordingly.

  1. Line 190: please change „…presented as mean and standard deviation“ into „…presented as means and standard deviations“

Response:  Changes were made accordingly.

  1. Line 204: maybe change „that stress“ into „the stress“

Response:  Changes were made accordingly.

  1. Line 251: maybe change „that in the present study…“ into „which in the present study…“

Response:  Changes were made accordingly.

  1. Line 356: maybe change „…reported that after….“ Into „…reported similar results after…“

Response:  Changes were made accordingly.

Reviewer 4 Report

The presented work is processed at a very good level and fits your journal with its focus. I have only small comments on the methodology of the experiment, I would recommend in this section to state in more detail the conditions under which the animals were kept in the farm.

I also recommend that you describe the conditions under which the calves were transported to the slaughterhouse, note how much space was provided to them during transport by means of transport and at what temperatures they were transported.

Next, I recommend stating how long the time interval passed from the arrival at the slaughterhouse to the last saliva collection.

Instead of the name of  the method Reactive oxygen species (ROS) I recommend to use the name 2'-7'-dichlorofluorescein diacetate staining.

Author Response

Responses to the comments of Reviewer #4

  1. I would recommend in this section to state in more detail the conditions under which the animals were kept in the farm.

Response:  The authors appreciate the question raised by the reviewer and details about this subject were included in the revised manuscript. The authors are grateful for the suggestion made by adding an indication of the charge density.

  1. I also recommend that you describe the conditions under which the calves were transported to the slaughterhouse, note how much space was provided to them during transport by means of transport and at what temperatures they were transported.

Response:  The authors are grateful for the recommendations. Weather conditions and the loading density were included in the revised manuscript.  

  1. Next, I recommend stating how long the time interval passed from the arrival at the slaughterhouse to the last saliva collection.

Response:  The authors are grateful for the recommendations. The resting period or lairage time was of a range between 15 and 17 hours. This information was included in the revised manuscript.  

  1. Instead of the name of the method Reactive oxygen species (ROS) I recommend to use the name 2'-7'-dichlorofluorescein diacetate staining.

Response:  Changes were made accordingly.